# Naturally Occurring Flavonoids and Isoflavonoids and Their Microbial Transformation: A Review

**DOI:** 10.3390/molecules25215112

**Published:** 2020-11-03

**Authors:** Jun-Fei Wang, Si-Si Liu, Zhi-Qiang Song, Tang-Chang Xu, Chuan-Sheng Liu, Ya-Ge Hou, Rong Huang, Shao-Hua Wu

**Affiliations:** 1Yunnan Institute of Microbiology, School of Life Sciences, Yunnan University, Kunming 650091, China; wang__junfei@163.com (J.-F.W.); liusisi1994@126.com (S.-S.L.); songzhiqiang1996@126.com (Z.-Q.S.); tcxu123@126.com (T.-C.X.); liucs313@126.com (C.-S.L.); houyage@126.com (Y.-G.H.); 2School of Chemical Science and Technology, Yunnan University, Kunming 650091, China; huangrong@ynu.edu.cn

**Keywords:** flavonoids, isoflavonoids, fungi, actinomycetes, biotransformation, biosynthesis

## Abstract

Flavonoids and isoflavonoids are polyphenolic secondary metabolites usually produced by plants adapting to changing ecological environments over a long period of time. Therefore, their biosynthesis pathways are considered as the most distinctive natural product pathway in plants. Seemingly, the flavonoids and isoflavones from fungi and actinomycetes have been relatively overlooked. In this review, we summarized and classified the isoflavones and flavonoids derived from fungi and actinomycetes and described their biological activities. Increasing attention has been paid to bioactive substances derived from microorganism whole-cell biotransformation. Additionally, we described the utilization of isoflavones and flavonoids as substrates by fungi and actinomycetes for biotransformation through hydroxylation, methylation, halogenation, glycosylation, dehydrogenation, cyclisation, and hydrogenation reactions to obtain rare and highly active biofunctional derivatives. Overall, among all microorganisms, actinomycetes are the main producers of flavonoids. In our review, we also summarized the functional genes involved in flavonoid biosynthesis.

## 1. Introduction

Flavonoids and isoflavonoids are versatile natural compounds and subdivisions of polyphenols that represent a large proportion of secondary metabolites produced by higher plants and are a rich part of the human diet [1]. They are playing multiple roles in the physiology and ecology of individual plant species. Flavonoids are a type of yellow pigment derived from 2-phenyl chromogenone as the parent nucleus and a series of compounds with C6-C3-C6 as the basic skeleton. Isoflavonoids are a subclass of flavonoids characterized by possessing a benzene-ring connected to C-3 instead of C-2 [2]. Phenylpropanoid and polyketone compounds are normally catalyzed by chalcone synthase to produce chalcones, and then cyclization of chalcones leads to generate flavonoids. Isoflavonoids, originating from the same biochemical pathway as flavonoids, are derived by aryl migration in a 2-phenylchroman skeleton under the catalysis of 2-hydroxyisoflavanone synthase [3,4,5,6]. Flavonoids and isoflavonoids are well-known natural products with extensive pharmacological activities and extremely low toxicity and therefore have become the focus and hotspots of drug discovery and development [7,8,9]. Most plants contain isoflavonoids and flavonoids that play important roles in the growth and development of plants as well as in antibacterial and disease-preventing aspects [10,11]. More importantly, they possess a wide range of biological activities, such as antibacterial, antifungal, antiviral [12,13], antitumour [8], anti-inflammatory [14], and antiaging activities [15].

Most research on isoflavonoids and flavonoids has focused on plant sources, and relatively less attention has been paid on microorganisms as the potential sources of these compounds. The presence of metabolic pathways of isoflavonoid and flavonoid biosynthesis in microorganisms has been confirmed. In particular, the presence of isoflavonoids and flavonoids in fungi and actinomycetes has been widely reported in studies [16,17]. Both fungi and actinomycetes exhibit unique and unusual biochemical pathways, and many important drugs have been derived from their secondary metabolites, for instance, penicillin, cyclosporin, paclitaxel, and statins are derived from fungi [18], and streptomycin, kanamycin, nystatin, and rifamycin are derived from actinomycetes [19,20]. Therefore, isoflavonoids and flavonoids derived from fungi and actinomycetes are of great significance. Fungi and actinomycetes are also the key biological sources of isoflavonoid and flavonoid biotransformation [21,22]. Microbial transformation is advantageous because of mild reaction conditions and high regio- and enantioselectivity. The literature suggests that almost all isoflavonoids and flavonoids derived from actinomycetes are produced by *Streptomyces* because the genes in this genus encode key enzymes such as phenylalanine ammonia-lyase and chalcone synthase (CHS) that catalyze isoflavone and flavone synthesis [23,24].

## 2. Isoflavonoids and Flavonoids from Actinomycetes and Fungi

### 2.1. Isoflavonoids

Prior to 2000, plant-derived natural products of isoflavones were mainly confined to the family Leguminosae [25,26]. Since then, several comprehensive reviews on the occurrence of isoflavones in non-leguminous plants have been published [27,28,29]. Recent reviews have focused on newly discovered isoflavonoids in legumes [30,31]. However, only a few of these compounds have been isolated from fungi and actinomycetes. Here, we classified the isoflavones isolated from fungi and actinomycetes into three categories, namely simple isoflavones, isoflavone glycosides, and complex isoflavones; the structures of the compounds are shown in Figure 1, Figure 2 and Figure 3, respectively. These compounds mainly exhibit antioxidant, anti-tumour, antimicrobial, and β-galactosidase inhibitory activities.

#### 2.1.1. Simple Isoflavonoids

Four known isoflavonoids, daidzein (**1**), genistein (**2**), daidzein-7-*O*-α-L-rhamnoside (**23**), and genistein-7-*O*-α-L-rhamnoside (**25**) were derived from Indonesian actinomycete *Streptomyces* sp. TPU1401A. Compounds **1** and **2** could be transformed into 7-*O*-glycosides, daidzein-7-*O*-α-L-rhamnoside and genistein-7-*O*-α-L-rhamnoside by strain TPU1401A, respectively. The strain TPU1401A grew better when adding daidzein-7-*O*-α-L-rhamnoside (**23**) and genistein-7-*O*-α-L-rhamnoside (**25**) instead of compounds **1** and **2**. This phenomenon suggested that the strain TPU1401A transformed isoflavones into isoflavone glycosides to hasten growth [32]. 

Five known isoflavones, genistein (**2**), prunetin (**3**), kakkatin (**4**), isoformononetin (**5**), and formononetin (**6**) were isolated from lichen-associated *Amycolatopsis* sp. YIM 130642. These compounds exhibited weak antimicrobial activities against several pathogens [33]. EtOH extracts of the fungal strain, *Aspergillus* sp. HK-388 afforded 8-hydroxydaidzein (**7**), which exhibited non-competitive inhibition on aldose reductase of human recombinant with its *K*_i_ value of 7.0 μM [34].

During the chemical screening of a number of actinomycete strains from marine and terrestrial sources, six isoflavones were obtained from several *Streptomyces* strains and identified to be daidzein (**1**), genistein (**2**), 7-*O*-methylgenistein (**3**), 4′,6-dihydroxy-7-methoxyisoflavone (**4**), 4′-hydroxy-6,7-dimethoxyisoflavone (**8**), 4′,7-bis-(β-cymaropyranosyl)-genistein (**28**), and genistein-4′-(6″-methyl)-salicylate (**49**), in which compounds **8**, **28**, and **49** are new compounds [35]. 

Three isoflavonoids, 4′,7,8-trihydroxyisoflavone (**7**), 3′,4′,7-trihydroxyisoflavone (**9**) and 8-chloro-3′,4′,5,7-tetrahydroxyisoflavone (**10**) were obtained from the culture broth of *Streptomyces* sp. OH-1049. These compounds showed antioxidant activity in vitro [36]. Compounds **9** and **10** were acetylated by adding pyridine and Ac_2_O to afford 3′,4′,7-triacetoxyisoflavone and 8-chloro-3′,4′,5,7-tetraacetoxyisoflavone, respectively [37].

Three new isoflavones, 3′,5,7-trihydroxy-4′,6-dimethoxyisoflavone (**11**), 3′,5,7-trihydroxy-4′,8- dimethoxyiso-flavone (**12**) and 3′,8-dihydroxy-4′,6,7-trimethoxyisoflavone (**13**), were derived from *Streptomyces* culture filtrates. Compounds **11** and **12** were able to suppress the catechol-*O*-methyltransferase and dopa decarboxylase, and exhibit hypotensive action. Compound **13** was a specific inhibitor of catechol-*O*-methyltransferase [38]. A novel isoflavone, 3′,4′,5,7-tetrahydroxy-8-methoxy (**14**), and four known compounds, genistein (**2**), psi-tectorigenin (**15**), orobol (**16**), and 8-hydroxygenistein (**17**) were obtained from the culture filtrates of fungi and streptomyces during the screening of inhibiting dopa decarboxylase. Compounds **14** and **16** exhibited remarkable activity in restraining dopa decarboxylase. Compound **16** also showed an excellent hypotensive effect on spontaneously hypertensive rats [39]. 

Four isoflavone glycosides and two isoflavones were isolated from *Streptomyces xanthophaeus*. They were determined as daidzein (**1**), genistein (**2**), daidzein-4′,7-di-α-L-rhamnoside (**22**), daidzein-7-α-L-rhamnoside (**23**), genistein-4′,7-di-α-L-rhamnoside (**24**), and genistein 7-α-L-rhamnoside (**25**). All of these compounds showed β-galactosidase inhibiting activities [40]. Ten isoflavone glycosides and one isoflavone, daidzein (**1**), were obtained from *Streptomyces* sp. RB1 associated with *Macrotermes natalensis* [17]. Later, seven isoflavone glycosides and one isoflavone, genistein (**2**) were additionally isolated from this strain [41]. Those glycosyl compounds are introduced in the following classification (Section 2.1.2). Eight known isoflavones, daidzein (**1**), genistein (**2**), 7-*O*-methyl genistein (**3**), kakkatin (**4**), 8-chlorogenistein (**19**), genistin (**33**), daidzin (**34**), glycitin (**37**), and a novel isoflavone named 7-*O*-methyl-8-chlorogenistein (**18**) were identified from *Streptomyces* strain YIM GS3536. The MIC values of **18** against *C*. *albicans*, *E*. *coli*, *B*. *subtilis*, and *S*. *aureus* were in the range of 23–35 μg/mL. Compound **18** also showed appreciable cytotoxicity against human leukemia cell lines (HL60) and human melanoma cell lines (B16) with IC_50_ values of 19.9 and 17.5 μg/mL, respectively [42]. 

#### 2.1.2. Isoflavonoid Glycosides

Three novel isoflavonoid glycosides, daidzein-4′-(2-deoxy-α-L-fucopyranoside) (**20**), daidzein-7-(2-deoxy-α-L-fucopyranoside) (**21**) and daidzein-4′,7-di-(2-deoxy-α-L-fucopyranoside) (**71**) were obtained from the culture broth of mangrove-derived actinomycete *Micromonospora aurantiaca* 110B. Though the new compounds were unable to inhibit pathogenic fungus or bacteria, they exhibited moderate cytotoxic activities to the human colon tumor cell line HCT116, the human lung carcinoma cell line A549, and hepatocellular liver carcinoma cell line HepG2 [43]. Four isoflavone glycosides were obtained from *S. xanthophaeus*. They were identified as daidzein-4′,7-di-α-L-rhamnoside (**22**), daidzein-7-α-L-rhamnoside (**23**), genistein-4′,7-di-α-L- rhamnoside (**24**), and genistein-7-α-L-rhamnoside (**25**). All of these compounds had β-galactosidase inhibiting activities [40].

From the fermentation broth of the lichen-associated *Amycolatopsis* sp. YIM 130642, a new isoflavonoid glycoside, 7-*O*-methyl-5-*O*-α-L-rhamnopyranosylgenistein (**26**), along with a firstly natural occurring isoflavonoid glycoside, 7-*O*-α-D-arabinofuranosyl daidzein (**27**) was isolated. Compound **26** showed moderate inhibitory activity towards *E. coli* and *S. aureus* with MIC value of 64 μg/mL [33]. A new isoflavone glycoside, 4′,7-bis-(β-cymaropyranosyl)-genistein (**28**) was isolated from *Streptomyces* sp. HKI 129-L [35].

Ten isoflavonoid glycosides were isolated from *Streptomyces* sp. RB1 originating from *M. natalensis*, including three novel ones, termisoflavones A−C (**29**–**31**) and seven known compounds, daidzein-4′,7-di-α-L-rhamnoside (**22**), daidzein-7-α-L-rhamnoside (**23**), genistein-4′,7-di-α-L- rhamnoside (**24**), genistein-7-α-L-rhamnoside (**25**), 6-*O*-methyl-7-*O*-α-L-rhamnopyranosyldaidzein (**32**), genistin (**33**), and daidzin (**34**). Unfortunately, these compounds showed no antimicrobial activities. Only compounds **23** and **34** showed protective effects on kidney cells [17].

One new isoflavonoid glycoside, termisoflavone D (**35**), together with six known analogues, daidzein-4′,7-di-α-L-rhamnopyranoside (**22**), daidzein-7-α-L-rhamnopyranoside (**23**), genistein-4′,7-di-α-L-rhamnopyranoside (**24**), genistein-7-α-D-glucopyranoside (**33**), daidzein-7-α-D-glucopyranoside (**34**), and 4′-*O*-methyl-7-*O*-α-L-rhamnopyranosylgenistein (**36**), were identified from *Streptomyces* sp. RB1. Compound **23** prevented glutamate-induced HT22 cell death by blocking the accumulation of intracellular reactive oxygen species (ROS) [41]. Three isoflavone glycosides, genistin (**33**), daidzin (**34**), and glycitin (**37**), were derived from *Streptomyces* strain YIM GS3536 [42]. Daidzein-7-*O*-α-L-rhamnoside (**23**) and genistein-7-*O*-α-L-rhamnoside (**25**) were derived from Indonesian actinomycete *Streptomyces* sp. TPU1401A [32].

#### 2.1.3. Complex Isoflavones

An antifungal luteone, 5,7,2′,4′-tetrahydroxy-6-(3,3-dimethylallyl) isoflavone (**38**), was obtained from the cultures of *Aspergillus flavus* and *Botrytis cinerea*. At the level of 7–14 μg/cm^2^, the growth of *Chlamydia herbarum* was completely inhibited by compound **38** [44]. Four compounds, luteone hydrate (**39**), luteone metabolite AF-2 (**40**), BC-1 (**41**), and BC-2 (**42**), were obtained from the co-cultures of *A. flavus* and *B. cinerea*. Five isoflavonoids, M-1-1 (**44**), M-1-2 (**45**)**,** M-2 (**46**), M-3-1 (**47**), and M-3-2 (**48**) were isolated from the fungus *A. flavus* or *B*. *cinereal* by converting licoisoflavone A (**43**). When their inhibitory ability to grow *Cladosporium herbarum* was tested, compounds **38** and **43** were found to completely inhibit growth at the level of 25–100 μg/cm^2^, while compound **44** had only a small inhibition zone at the level of 100 μg/cm^2^ [45]. A new isoflavone, genistein-4′-(6″-methyl)-salicylate (**49**) was obtained from *Streptomyces* sp. GW27/2506 [35].

### 2.2. Flavonoids

Flavonoid derivatives are natural products commonly found in medicinal plants and are synthesised from phenylpropanoid and acetate-derived precursors [11]. Some flavonoids have also been discovered from fungi and actinomycetes. We also classified these flavonoids into three groups: simple flavonoids, flavonoid glycosides, and complex flavonoids, and the structures of the compounds are shown in Figure 4, Figure 5 and Figure 6, respectively. Most of these compounds mainly exhibit antioxidant, antimicrobial, anti-inflammatory, antitumour, antifouling, and α-glucosidase inhibitory activity.

#### 2.2.1. Simple Flavonoids

*Trichoderma* strains are widely used to inhibit pathogens and promote the growth of plants. LC-QQQ-MS showed that three simple flavonoids, dihydromyricetin (**50**), isorhamnetin (**51**), and 4-hydroxy-5,7-dimethoxyflavanone (**52**), existed in the culture broth of fungus *T*. *asperellum* TJ01, and their proportion was the highest when cultured at 72 h. [46].

Three flavonoids, named 7-*O*-methylnaringenin (**53**), pinocembrin (**54**), and (−)-epicatechin (**55**), were isolated from the endophytic fungus *Annulohypoxylon elevatidiscus* BCRC 34014, which was isolated from decorticated woods. It is worth mentioning that this is the first time that flavonoids have been found in this species [47].

The endophytic fungus *Xylaria papulis* BCRC 09F0222 was isolated from hairy woody plants. Three flavonoids, named myricetin (**50**), myricitrin (**73**), and quercitrin (**74**), were obtained from this strain [48]. Since the fungus was cultivated on rice, which is a feeding plant known to produce myricetin and other flavonoids [49], these metabolites might be derived from rice or synthesized by modifying the existing flavonoid precursors in culture medium. Dechlorochlorflavonin (**56**), a known flavonoid derivative, was isolated from the fungus *Aspergillus candidus* Bdf-2 derived from insects. Compound **56** was assessed for antibacterial activities against *Ralstonia solanacearum* and *Staphylococcus aureus* ATCC29213 with MIC values of 64 µg/mL. However, no antioxidant activity was detected [50]. 

A new flavonoid, named 6,8,5′,6′-tetrahydroxy-3′-methylflavone (**57**), was obtained from the fungal strain *Penicillium* sp. SCSGAF 0023. This strain could help the host gorgonian corals against marine pathogenic bacteria, biological pollution organisms, and other invaders by producing antifouling and antibacterial compounds. Compound **57** had potent antifouling activity against *Balanus amphitrite* larvae settlement with an EC_50_ value of 6.7 μg/mL [51]. A new antifungal antibiotic, 3′-chloro-2′,5-dihydroxy-3,7,8-trimethoxy flavone (**58**) was isolated from *A. candidus* strains ATCC 20022 and CMI 16046 [52]. Two new flavonoids named aspergivones A (**59**) and B (**60**) were isolated from the fungus *A. candidus*. The strain was isolated from the gorgonian coral *Anthogorgia ochracea,* which was collected from the South China Sea. Compound **60** displayed only weak inhibitory activity against α-glucosidase with an IC_50_ value of 244 μg/mL. Cytotoxicity and antibacterial activity were not showed for compound **59** [53].

In addition, chlorflavonin (**58**), 3′-bromo-2′,5-dihydroxy-3,7,8-trimethoxyflavone (**61**), and dechlorochlorflavonin (**62**) were isolated from the fermentation broth of the fungus *Acanthostigmella* sp. CL12082. Compound **61** showed strong antifungal activity against *Aspergillus fumigatus* and *Candida albicans* with IC_50_ values of 0.54 and 0.11 μg/mL. Compound **58** displayed significant activity against the growth of pathogenic fungi, *C. albicans* and *A. fusigatus* with IC_50_ values of 0.035 and 0.10 μg/mL, as well as against the growth of HeLa cells IC_50_ values of 20 μg/mL. At the same time, these three compounds showed weak inhibition of growth of pathogenic fungi, *Cryptococus neoformans* with IC_50_ values of 20, 12, and 16 μg/mL, respectively [54]. Quercetin (**63**) was produced by an endophytic fungi *Psathyrella candolleana* from the seed of Ginkgo biloba. It displayed antibacterial activity against *S. aureus* with MIC values of 0.3906 mg/mL [55].

A study reported the isolation and determination of WS7528 (**64**), produced by *Streptomyces* sp. No. 7528 derived from a soil sample obtained at Nara Prefecture, Japan. It was tested orally and subcutaneously in immature rats to verify its effect on the growth of the uterus, which had also weak anti-inflammatory activity and could induce growth of the cell line MCF-7 [56]. Two flavonoids, rhamnazin (**65**) and cirsimaritin (**66**), obtained from microbial sources for the first time, were obtained from protoplast fusion between *Streptomyces* strains Merv 1996 and Merv 7409. Compound **65** showed remarkable activities against filamentous fungi *B. fabae* and *A. niger* with concentrations of 2.5 and 1.0 μg/mL, respectively. Compound **66** showed a strong antifungal activity in vitro against *C. neoformans*, *C. albicans*, *Pichia angusta*, and *Rhodotorula minuta* with MIC value of 1 μg/mL, while showed no inhibitory effect against filamentous fungi [57].

#### 2.2.2. Flavonoid Glycosides

Flavonoid glycosides derived from endophytic fungi were reported for the first time in 2016. Three quercetin-3-*O*-glycosides, named guaijaverin (**67**), isoquercitrin (**68**), and hyperin (**69**), were isolated from the endophytic fungus *Nigrospora oryzae*. The fungus was isolated from the leaves of the Nigerian mistletoe *Loranthus micranthus*, which was widely used in African traditional medicine [16]. Actinoflavoside (**70**), possessing a rare 2,3,6-trideoxy-3-amino-ribopyranoside aminosugar skeleton, was obtained from the fermentation broth of a marine *Streptomyces* sp. [58]. A flavonoid glycoside, daidzein-4′,7-di-(2-deoxy-α-L-fucopyranoside) (**71**), was obtained from actinomycetes, which showed moderate activity against tumor cells of A549, HepG2, and HCT116 [43].

The strain *Streptomyces* sp. ERINLG-4 was isolated from the soil samples collected from a depth of 5–15 cm in the Doddabetta forest. The EtOAc extract of the strain showed potent cytotoxic activity in vitro against the A549 lung adenocarcinoma cancer cell line. The following work led to the isolation of an active component, quercetin-3-*O*-β-L-rhamnopyranosyl-(1→6)-β-D-glucopyranoside (**72**), showing prominent cytotoxic activity against the A549 lung cancer cell line with an IC_50_ value of 82 μg/mL. However, it showed no toxicity against the Vero normal cell line, up to 2000 μg/mL [59]. Three flavonoids, named myricetin (**50**), myricitrin (**73**), and quercitrin (**74**) were obtained from the endophytic fungus *X. papulis* BCRC 09F0222 isolated from hairy woody plants [48].

#### 2.2.3. Complex Flavonoids

Two new lavandulylated flavonoids, 6-lavandulyl-7-methoxy-5,2′,4′-trihydroxylflavanone (**75**) and 5′-lavandulyl-4′-methoxy-2,4,2′,6′-tetrahydroxylchalcone (**76**) were isolated from the sponge-derived actinomycete *Streptomyces* sp. G246. Compounds **75** and **76** showed a broad spectrum of antimicrobial activity. Compound **76** showed excellent inhibitory effects on *C. albicans*, *S. aureus*, *Bacillus cereus*, *Enterococcus faecalis*, *Salmonella enterica*, and *Pseudomonas aeruginosa*, but no inhibitory effect on *Escherichia coli*. [60]. 

Three new lavandulylated flavonoids, (2S,2″S)-6-lavandulyl-7,4′-dimethoxy-5,2′- dihydroxylflavanone (**77**), (2S,2″S)-6-lavandulyl-5,7,2′,4′-tetrahydroxylflavanone (**78**) and (2″S)-5′-lavandulyl-2′-methoxy-2,4,4′,6′-tetrahydroxylchalcone (**79**), together with two known compounds, (2S,2″S)-6-lavandulyl-7-methoxy-5,2′,4′-trihydroxyl-flavanone (**80**) and 6-prenyl-4′-methoxy-5,7-dihydroxylflavanone (**81**), were isolated from the fermentation broth of *Streptomyces* sp. G248. Compounds **77**–**79** showed remarkable antimicrobial activity, and compounds **80** and **81** were found to inhibit *Mycobacterium tuberculosis* H37Rv with the MIC values of 6.0 and 11.1 µg/mL [61]. A strain of *Penicillium griseoroseum* was isolated from *Coffeea arabica* seeds, and its metabolites were studied in 5,7,3′,4′,5′-pentamethoxyflavanone supplemented medium to obtain bezylated flavanone (**82**) [62].

## 3. Microbiological Transformation of Isoflavonoids and Flavonoids

Biotransformation is a well-known process of effectively obtaining isoflavones and flavonoids. The main reactions in the conversion process include hydroxylation, methylation, glycosylation, and cyclisation, through which some of the rare or expensive isoflavones and flavonoids can be obtained [63,64]. In this section, we review the biotransformation of isoflavones and flavonoids by fungi and actinomycetes, and the compounds involved in the microbiological transformation are shown in Figure 7 and Figure 8, respectively.

### 3.1. Biotransformation of Isoflavones and Flavonoids by Fungi

In order to obtain *O*-dihydroxyisoflavones, *Aspergillus saitoi* was used for transforming daidzein (**1**) and genistin (**33**) to generate 8-hydroxydaidzein (**7**) and 8-hydroxygenistein (**17**), respectively, by deglycosylation and hydroxylation [65]. 

Three isoflavones were used to study the biotransformation by *Aspergillus niger*, and it was found that daidzein (**1**) was not metabolized. The C-6 position of 7,4′-dimethoxyisoflavone (**83**) was hydroxylated to form 6-hydroxy-7,4′-dimethoxyisoflavone (**85**), which was then converted into **1** by demethylation of C-7 and C-4′ positions. In addition, 7,4′-diacetoxyisoflavone (**84**) was converted into **1** through the hydrolysis of C-7 and C-4′ positions [66]. 6,7,4′-Trimethoxyisoflavone (**86**) and 5,7,4′-trimethoxyisoflavone (**87**) were studied for their biotransformation by *A. niger*. They could be converted to 4′-hydroxy-6,7-dimethoxyisoflavone (**8**) and 4′-hydroxy-5,7- dimethoxyisoflavone (**88**), respectively, by demethylation at C-4′ position [67].

The isoflavone, luteone [5,7,2′,4′-tetrahydroxy-6-(3,3-dimethylallyl) isoflavone] (**38**) could be transformed into 2″,3″-dihydro-3″-hydroxyluteone (**89**), 2″,3″-dihydrodihydroxyluteone (**90**), dihydrofuranoisoflavone (**41**) and dihydropyranoisoflavone (**91**) by cultures of *A. flavus* and *B. cinerea*. The main products **41** and **89** were less toxic to *C. herbarum* than compound **38** [44]. Two isoflavones, 7-*O*-methyl-2,3-dehydrokievitone epoxide (**93**) and 7-*O*-methyl-2,3-dehydrokievitone glycol (**94**), were formed from 2,3-dehydrokievitone (**92**) by methylating with ethereal diazomethane under the metabolism by *B. cinerea*. The epoxide was regarded as the key metabolic intermediate involved in the transformation from precursor compounds containing a prenyl side-chain with ortho-hydroxylation to 2,3-dihydroxy-3-methylbutyl-substituted isoflavones [68].

Biotransformation of two flavonoids was studied by using four strains of *A. niger*, three of which were mutated by ultraviolet irradiation. Flavanone (**95**) and 6-hydroxyflavanone (**96**) were transformed to flavan-4-ol (**99**) and 6-hydroxyflavan-4-ol (**101**), respectively, through decarbonylation, and the dehydrogenation at C-2 and C-3 positions produced flavone (**97**) and 6-hydroxyflavone (**102**), respectively. Compound **95** was reduced at C-4 and hydroxylated at C-7 to form 7-hydroxyflavan-4-ol (**100**)**,** and dehydrogenation at C-2 and C-3, along with hydroxylation at C-3 to from 3-hydroxyflavone (**98**) [69].

2″,3″-Dihydro-3″-hydroxywighteone (**106**) was obtained by biotransformation from the antifungal compound, wighteone [5,7,4′-trihydroxy-6-(3,3-dimethylallyl) isoflavone] (**103**) under the modification of *A. flavus* and *B. cinerea* and the latter was further transformed into dihydrofurano-isoflavone (**104**). Small amounts of dihydropyrano-isoflavone (**105**), dihydrofurano-isoflavone (**104**), and 2″,3″-dihydrodihydroxywighteone (**107**) were obtained in *A. flavus* culture medium. Compounds **105** and **107** were obtained in *B. cinerea* medium [70].

The study on the biotransformation of filamentous fungi found that *Trichoderma harzianum* NJ01 could convert puerarin (**108**) to 3′-hydroxypuerarin (**109**) with a conversion rate of up to 41% under the optimal conditions. In the DPPH free radical scavenging system, compound **109** was 20 times more active than **108**. The solubility of **109** is 1.3 times higher than **108** [71]. Glabratephrin (**110**) was obtained from *Tephrosia purpurea* and transformed into pseudosemiglabrin (**111**) by the culture of *A. niger*. The transformation process is realized by using the open loop and the closed loop of the five-member ring [72]. 

Flavanone (**95**) is converted by *A. niger* MB and *Penicillium chermesinum* 113, and two flavonoids, 6-hydroxyflavanone (**96**) and 4-hydroxyflavanone (**114**), were obtained by hydroxylation, and three dihydrochalcones with hydroxyl groups were formed and identified as 2′,5′-dihydroxydihydrochalcone (**112**), 2′-hydroxydihydrochalcone (**113**), and 2′,4-dihydroxydihydrochalcone (**115**), respectively [21].

Prenylated flavonoids are usually sourced from medicinal plants, and because of restrictions on bioavailability and the high extraction costs, it is essential to study their biological origins. Studies on the biotransformation showed that naringenin 8-dimethylallyltransferase expressed by transgenic yeast could convert naringenin (**116**) into 8-dimethylallylnaringenin (**117**). This was an example to provide a method for the production of prenylated flavonoids that rarely occur in nature [73].

Biotransformation by *A. niger* MB using 6- and 7-methoxyflavones as substrates showed that 6-methoxyflavone (**118**) and 7-methoxyflavone (**120**) were converted to 6-hydroxyflavone (**102**) and 7-hydroxyflavone (**121**) by demethylation at C-6, and then to from 6,4′-dihydroxyflavone (**119**) and 7,4′-dihydroxyflavone (**122**) by demethylation at C-6 and hydroxylation at C-4′ [74]. The resulting products had a stronger antioxidant activity than the substrates.

The biotransformation of 7-hydroxyflavanone (**123**) by three *Aspergillus* strains (*A. ochraceus* 456, *A. niger KB, A. niger* 13/5) and one *Penicillium* strain (*P. chermesinum* 113) were studied. In *P. chermesinum* 113, compound **123** was converted to 7-methoxyflavanone (**124**) by methylation with a conversion rate of 24%, and to obtain 3′,4′-dihydroxy-7-methoxyflavanone (**125**) by methylation combined with hydroxylation at C-3 and C-4 positions, with a conversion rate of 19%. In *A. ochraceus* 456 and *A. niger KB*, the carbonyl group of compound **123** was reduced to form 2,4-*cis*-7-hydroxyflavan-4-ol (**100**) and compound **125**, respectively, with each conversion yields of 74% and 12%. 2,4-*trans*-5,7-Dihydroxyflavan-4-ol (**126**) was obtained by *A. ochraceus* transformation using compound **123** as substrate. In *A. niger* 13/5, substrate **123** was converted to 7-hydroxyflavone (**121**) by dehydrogenation at C-2 and C-3 positions with a conversion rate of up to 98%. Most of the products were found to have more oxidative activity than the substrate [75].

Xanthohumol (**127**) was converted as a substrate to obtain (2*S*)-8-[4″-hydroxy-3″-methyl- (2″-*Z*)-butenyl]-4′,7-dihydroxy-5-methoxy-flavanone (**130**) and (2*S*)-8-[5″-hydroxy-3″-methyl-(2″-*E*)- butenyl]-4′,7-dihydroxy-5-methoxyflavanone (**131**) by *Cunninghamella echinulata* NRRL 3655. Compounds **127**, **130**, (*E*)-2″-(2″′-hydroxyisopropyl)-dihydrofurano[2″,3″:4′,3′]-2′,4-hydroxy-6′- methoxychalcone (**128**) and (2*S*)-2″-(2‴-hydroxyisopropyl)-dihydrofurano[2″,3″:7,8]-4′-hydroxy- 5-methoxyflavanone (**129**) were obtained from *Pichia membranifaciens* which showed antimalarial activity against *Plasmodium falciparum* [76].

Three compounds, apigenin 5-*O*-α-L-rhamnopyranosyl-(1→3)-β-D-glucopyranoside (**132**), 5-*O*-α-L-rhamnopyranosyl-(1→2)-(6″-*O*-acetyl)-β-D-glucopyranoside (**133**) and chrysoeriol 5-*O*-α-L-rhamnopyranosyl- (1→4)-(6″-*O*-acetyl)-β-D-glucopyranoside (**134**) were obtained from the leaves of *Cephalotaxus harringtonia*, and they were used as substrates for biotransformation by *Paraconiothyrium variabile*. Compounds **133** and **134** were deglycosylated to form apigenin (**135**), then compound **134** was converted to chrysoeriol (**136**) [77].

Biotransformation was performed using isoflavones and 4′-fluoroisoflavones as substrates by *A. niger* and *Cunninghamella elegans*. Both fungi rapidly convert isoflavones into several metabolites. They metabolize isoflavones (40 mg/L) with half-lives of 1.6 and 4.2 days, respectively. Twenty-three metabolites were preliminarily identified during the biotransformation of *A. niger*. In the early stage, the main metabolites were mono-hydroxyl and dihydroxyl isoflavones, and after 10 days, the main metabolites were dihydroxyl and trihydroxyl isoflavones. The hydroxylation of isoflavones usually occurs in the B ring. Among them, 3′,4′-dihydroxyl analogues were the most abundant. Methoxy metabolites accumulate slowly during culture. In addition, some glycosides have been detected. However, 4′-fluoroisoflavones were not transformed during culture, indicating that there was regional selective hydroxylation in the initial metabolism of isoflavones [78].

α-Naphthoflavone (7,8-benzoflavone) (**137**) and β-naphthoflavone (5,6-benzoflavone) (**139**) were transformed by *Aspergillus glaucus* AM 211, *A. niger* UPF 724, *Penicillium thomi* AM 91, *Cladosporium avellaneum* AM 135, and *Verticillium* sp. AM 424 via *O*-demethylation and hydroxylation. The substrate α-NF was metabolized in seven days to produce 4′-hydroxy-α-naphthoflavone (**141**) and 4′-hydroxy-β-naphthoflavone (**142**). The most effective biocatalysts were two strains of AM 211 and AM 424, which converted **137** to **141** with the rates of 43% and 63%, respectively. Among them, only three strains AM 135, AM 424, and AM 211 can transform β-NF (**139**) into compounds **141** and **142**. Besides, 4′-methoxy-α-NF (**138**), a derivative of naphtoflavone, was transformed to compound **141** by *A*. *niger* UPF 724 and *P*. *thomi* AM 91, and the derivative 4′-methoxy-β-NF (**140**) was transformed to compound **142** by *A*. *niger* UPF 724 [79].

Bavachinin (**143**) was transformed in the broth of *Cunninghamella blakesleeana* AS 3.0970 to 4″-hydroxybavachinin (**146**) and 2″,3″-dihydroxybavachinin (**147**), which were obtained at the yields of 18% and 7%, respectively, after five days of biotransformation. Biotransformation of bavachin (**144**) by using *C. blakesleeana* AS 3.0910 cultures resulted in the formation of three products: (2R,2S)-bavachin 7-*O*-β-D-glucopyranoside epimers (**148**), (2*S*)-4″-hydroxybavachin (**149**) and (2*S*)-5″-hydroxybavachin (**150**), which were isolated in the yields of 26%, 2%, and 5%, after five days of biotransformation. In addition, *C. blakesleeana* AS 3.0910 transformed isobavachalcone (**145**) to isobavachalcone 4-*O*-β-D-glucopyranoside (**151**), which was obtained in 23% yields, after 5 days of biotransformation [80]. 6-Methylflavone (**152**) was transformed in the broth of entomopathogenic fungus *Isaria fumosorosea* KCH J2 to 6-methylflavone 8-*O*-β-D-(4″-*O*-methyl)-glucopyranoside (**153**) and 6-methylflavone 4′-*O*-β-D-(4″-*O*-methyl)-glucopyranoside (**154**), which were obtained in 60.5% and 39.5% yields, respectively, after 7 days of biotransformation [81].

*Isaria* filamentous fungi are often used as effective biocatalysts for the transformation of flavonoids. Flavone (**97**) was metabolized by *I. fumosorosea* KCH J2 in 5 days to produce flavone 2′-*O*-β-D-(4″-*O*-methyl)-glucopyranoside (**155**), flavone 4′-*O*-β-D-(4′-*O*-methyl)-glucopyranoside (**156**) and 3′-hydroxyflavone 4′-*O*-β-D-(4″-*O*-methyl)-glucopyranoside (**157**) with the yields of 5.5%, 14% and 4%, respectively. Compound **156** was also obtained when using *Isaria farinosa* J1.6 as biocatalyst. Besides, 5-hydroxyflavone (**158**) was converted into 5-hydroxyflavone 4′-*O*-β-D-(4″-*O*-methyl)-glucopyranoside (**159**) with 29% yield after 7 days in the culture of strain KCH J2. Compound **159** was also obtained under the biocatalysis of *I*. *farinosa* J1.4. Compound **102**, 6-hydroxyflavone, was metabolized by strain KCH J2 in 5 days to produce 6-*O*-β-D-(4″-*O*-methyl)- glucopyranoside (**160**) with 13% yield. Compound **160** was obtained when *I*. *farinosa* KW1.2 was used as a biocatalyst. 7-Hydroxyflavone (**121**) was metabolized by strain KCH J2 in 7 days to produce 7-*O*-β-D-(4″-*O*-methyl)-glucopyranoside (**161**) with 16% yield. Compound **161** was additionally obtained when both of the strains J1.4 and J1.6 were used as a biocatalyst. Daidzein (**1**) was metabolized by strain KCH J2 in 7 days to obtain 4′-hydroxyisoflavone 7-*O*-β-D-(4″-*O*-methyl)- glucopyranoside (4″-*O*-methyldaidzin) (**162**) with 14.8% yield. Compound **162** was also obtained when strains KW1.2 and J1.6 were used as biocatalyst. 7-Aminoflavone (**163**) was metabolized by strain KCH J2 in 7 days to obtain 7-acetamidoflavone (**164**) with 10% yield and 4′-hydroxy-7-acetamidoflavone (**165**) with 3% yield [82].

There is a similar report about *I. fumosorosea* KCH J2. 2′-Methoxyflavanone (**166**) was transformed in the broth of the strain KCH J2 to 2′-methoxyflavanone 5′-*O*-β-D-(4″-*O*-methyl)-glucopyranoside (**167**) with 18.6% yield and flavan-4-ol 2′-*O*-β-D-(4″-*O*-methyl)-glucopyranoside (**168**) with 24.9% yield after 11 days of biotransformation. 3′-Methoxyflavanone (**169**) was metabolized by strain KCH J2 in 7 days to produce flavan-4-ol 3′-*O*-β-D-(4″-*O*-methyl)-glucopyranoside (**170**) with a yield of 15.4% and 3′-hydroxyflavanone 6-*O*-β-D-(4″-*O*-methyl)-glucopyranoside (**171**) with a yield of 16%. 4′-Methoxyflavanone (**172**) was metabolized by strain KCH J2 in 7 days to produce 4′-*O*-β-D-(4″-*O*-methyl)-glucopyranoside (**173**), 4′-hydroxyflavanone 6-*O*-β-D-(4″-*O*-methyl)-glucopyranoside (**174**) and 3′,4′-dihydroxyflavanone 6-*O*-β-D-(4″-*O*-methyl)-glucopyranoside (**175**) with the yields of 11%, 20.7% and 32.8%, respectively. 6-Methoxyflavanone (**120**) was metabolized by strain KCH J2 in 8 days to produce 6-methoxyflavanone 4′-*O*-β-D-(4″-*O*-methyl)-glucopyranoside (**176**), and 3′-hydroxy-6- methoxyflavanone-4′-*O*-β-D-(4″-*O*-methyl)-glucopyranoside (**177**) with the yields of 6% and 7%, respectively. Compound **118** was metabolized by the strain KCH J2 in 12 days to produce 6-methoxyflavanone 3′-*O*-β-D-(4″-*O*-methyl)-glucopyranoside (**178**), 6-methoxyflavone 4′-*O*-β-D-(4″-*O*-methyl)-glucopyranoside (**179**), and 3′-hydroxy-6-methoxyflavone 4′-*O*-β-D-(4″-*O*- methyl)-glucopyranoside (**180**) with yields of 2.6%, 7.8%, and 7.9%, respectively [83].

When *I. fumosorosea* KCH J2 as the biocatalyst, the substrate 3-hydroxyflavone (**98**) was metabolized in 7 days to produce flavone 3-*O*-β-D-(4″-*O*-methyl)-glucopyranoside (**181**), flavone 3-*O*-β-D-glucopyranoside (**182**), and 3-*O*-[β-D-glucopyranosyl-(1→6)-β-D-glucopyranosyl]- 4′-hydroxyflavone (**183**) with the conversion rates of 42.5%, 5%, and 3.5%, respectively. Compounds **181** and **182** were formed by the biocatalysis of the strains J1.4 and J1.6. Besides, 3-methoxyflavone 4′-*O*-β-D-(4″-*O*-methyl)-glucopyranoside (**185**) was obtained with 29% yield from the transformation of 3-methoxyflavone (**184**) in the culture of strain KCH J2 for 7 days. Compound **185** was also obtained when strain J1.4 and J1.6 were used as biocatalysts. When the strain KCH J2 was cultivated for 10 days, 3′,4′,5,7-tetrahydroxyflavone 3-*O*-β-D-(4″-*O*-methyl)- glucopyranoside (**187**) and 3′,4′,5,7-tetrahydroxyflavone 3-*O*-β-D-glucopyranoside (isoquercetin) (**69**) were obtained by biotransformation of 3,3′,4′,5,7-pentahydroxyflavone (quercetin) (**186**) with yields of 18.5% and 12%, respectively. 5,6,7-Trihydroxyflavone (baicalein) (**188**) was transformed in the broth of the strain KCH J2 to 5,7-dihydroxyflavone 6-*O*-β-D-(4″-*O*-methyl)-glucopyranoside (**189**) with yield of 9.7% after 14 days [84].

Glycosylation is an effective means to regulate the solubility, stability, bioavailability, and bioactivity of natural product drugs. Several important flavonoid aglycones were studied for their glycosylated transformation by microorganisms. Naringenin (**116**) was glycosylated to form 4′,5-dihydroxyflavanone 7-*O*-β-D-(4-*O*-methyl) glucopyranoside (**190**), 5,7-dihydroxyflavanone 4′-*O*-β-D-(4-*O*-methyl) glucopyranoside (**191**), 5,7-dihydroxyflavanone 4′-*O*-β-D-glucopyranoside (**192**) and 5-hydroxyflavanone 4′,7-di-*O*-β-D-(4-*O*-methyl) glucopyranoside (**193**) by *I. fumosorosea* ACCC 37814. Luteolin (**194**) was transformed to 4′,5,7-trihydroxyflavone 3′-*O*-β-D-(4-*O*-methyl) glucopyranoside (**195**) and 3′,5,7- trihydroxyflavone 4′-*O*-β-D-(4-*O*-methyl) glucopyranoside (**196**) by the strain ACCC 37814 via (4-*O*-methyl)-glucosyl substituted. Diosmetin (**197**) was transformed to 5,7-dihydroxy-4′-methoxyflavone 3′-*O*-β-D-(4-*O*-methyl) glucopyranoside (**198**) by the strain ACCC 37814 via 4-*O*-methyl-glucosyl substituted. Formononetin (**6**) was glycosylated to generate 4′-methoxyisoflavone 7-*O*-β-D-(4-*O*-methyl) glucopyranoside (**199**) by the strain ACCC 37814 [85].

The host plant-derived flavonoid, kaempferol-*O*-glycoside (**203**), was glycosylated and acetylated to produce kaempferol 3-*O*-[α-rhamnopyranosyl-(1→6)-β-galactopyranoside] (**201**), flavonol kaempferol (**202**), and acetylated flavonoid glycoside (**200**) by the fungal endophyte *Epicoccum nigrum*. Among them, compound **200** was a new kaempferol-*O*-diglycoside [86].

2′-Methoxyflavone (**204**) was transformed by *I. fumosorosea* KCh J2 via demethylation and 4-*O*-methylglycosylation. The substrate 2′-methoxyflavone was metabolized in 10 days to produce 2′-*O*-β-D-(4″-*O*-methylglucopyranosyl)-flavone (**209**), 8-*O*-β-D-(4″-*O*-methylglucopyranosyl)-2′- methoxyflavone (**210**), 5′-*O*-β-D-(4″-*O*-methylglucopyranosyl)-2′-methoxyflavone (**211**) and 3-*O*-β-D-(4″-*O*-methylglucopyranosyl)-2′-methoxyflavone (**212**). Compound **204** was transformed to 2′-hydroxyflavone (**213**) by *Beauveria bassiana* KCh J1. Similar to the biotransformation of compound **204**, 3′-methoxyflavone (**205**) was converted to 3′-hydroxyflavone (**214**) by the strain KCh J1. Apart from this, the strain *I. farinosa* KCh KW 1.1 biotransformed 3′-methoxyflavone (**205**) to 3′-*O*-β-D-(4″-*O*-methylglucopyranosyl)-flavone (**215**) efficiently. The strain *B. bassiana* KCh J1.5 enabled to biotransform 4′-methoxyflavone (**206**) to 4′-hydroxyflavone (**216**). Besides, compound **206** was transformed to 4′-*O*-β-D-(4″-*O*-methylglucopyranosyl)-flavone (**217**) by *B. bassiana* KCh J1.5. 2′,5′-Dimethoxyflavone (**207**) was transformed to **211**, 2′-*O*-β-D-(4″-*O*-methylglucopyranosyl)-5′- methoxyflavone (**218**) and 4′-O-β-D-(4″-*O*-methylglucopyranosyl)-2′,5′-dimethoxyflavone (**219**) by *B. bassiana* KCh J1.5 and *I. farinosa* KCh KW1.1. Furthermore, **207** was transformed to 5′-hydroxy-2′-methoxyflavone (**220**) and 4′-hydroxy-2′,5′-dimethoxyflavone (**221**) by the culture of *B. bassiana* KCh J1. 3′,4′,5′-Trimethoxyflavone (**208**) was transformed to 3′-*O*-β-D-(4″-*O*-methylglucopyranosyl)-4′,5′-dimethoxyflavone (**222**), 4-*O*-methylglucoside (**223**), 6-*O*-β-D-(4″-*O*-methylglucopyranosyl)-3′,4′,5′-trimethoxyflavone (**224**) and 3′-*O*-β-D-(4″-*O*- methylglucopyranosyl)-6-hydroxy-4′,5′-dimethoxyflavone (**225**) by the culture of *B. bassiana* KCh J1.5 [87].

Six chalcones, 2″,4″-dimethoxy-2′-hydroxychalcone (**226**), 2″,3″-dimethoxy-2′-hydroxychalcone (**227**), 3″,4″-dimethoxy-2′-hydroxychalcone (**228**), 3″,5″-dimethoxy-2′-hydroxychalcone (**229**), 2″,3″,4′-trimethoxy-2′-hydroxychalcone (**230**) and 2″,4″,4′-trimethoxy-2′-hydroxychalcone (**231**) can be converted to seven modified chalcones and six flavonoids in the broth of fungus *A. niger* LSPN001, which passed through two different pathways. Among them, the modified chalcones included 2″-methoxy-2′,5″-dihydroxychalcone (**232**), 2″-methoxy-2′,3′,5″-trihydroxychalcone (**233**), 2″-methoxy-2′,3′-dihydroxychalcone (**234**), 2′,3″,4″-trihydroxychalcone (**235**), 5″-methoxy-2′,4″- dihydroxychalcone (**236**), 4′,2″-dimethoxy-2′,3″-dihydroxychalcone (**237**), 2′,4′,2″,4″- tetrahydroxychalcone (**238**). The flavanones chalcones included 2′,4′-dimethoxyflavanone (**239**), 2′-methoxy-4′-hydroxyflavanone (**240**), 4′,8-dihydroxy-2′-methoxyflavanone (**241**), 2′,3′-dimethoxyflavanone (**242**), 3′-hydroxy-2′-methoxyflavanone (**243**), 3′,4′-dimethoxyflavanone (**244**), 3′,4′-dihydroxyflavanone (**245**), 3′,5′-dimethoxyflavanone (**246)**, 3′-hydroxy-5′-methoxy flavanone (**247**), 7,2′,3′-trimethoxyflavanone (**248**), 3′-hydroxy-7,2′-dimethoxyflavanone (**249**), 7,2′,4′-trimethoxyflavanone (**250**), and 2′,4′-dihydroxy-7-methoxyflavanone (**251**) [88].

### 3.2. Biotransformation of Isoflavones and Flavonoids in Actinomycetes

5,7,4′-Trihydroxyisoflavon (**2**) is present in soybean meal. Fermentation media containing soybean meal could be used to isolate two new isoflavonoids, 8-chlorogenistein (**19**) and 6,8-dichlorogenistein (**252**), from *Streptomyces griseus*. However, the strain could not produce compounds **19** and **252** without soybean meal, and this suggests that the strain could biotransform compound **2** into chlorinated metabolites **19** and **252**. These new isoflavonoids were produced through microbial halogenation. When *S. griseus* was cultivated with radiologically labelled acetate or phenylalanine, no labelled isoflavones were obtained. The result illustrated that isoflavonoids isolated from streptomycetes may have originated from the medium containing plant-derived nutrient components rather than having a microbial biosynthetic origin [63].

A shuffled biphenyl dioxygenase holoenzyme with broad substrate specificity was coded by the bphA1(2072)A2A3A4 gene cluster, and the gene cluster was introduced into *Streptomyces lividans*. The recombinant *S. lividans* cell could transform flavone **(97**) and flavanone (**95**) into 3′-hydroxyflavone (**214**), 2′,3′-dihydroxyflavone (**253**), 2′,3′-dihydroxyflavanone (**254**), 3′-hydroxyflavanone (**255**), and 2′-hydroxyflavanone (**256**). In addition, 6-hydroxyflavone (**102**) and 6-hydroxyflavanone (**96**) were transformed into 3′,6-dihydroxyflavone (**257**) and 2′,6-dihydroxyflavanone (**258**), respectively. Among all the biotransformed compounds, only compounds **253** and **254** exhibited free radical scavenging activity [89].

Two prenyltransferases, NphB and SCO7190, obtained from two *Streptomyces* sp., namely CL190 and *S. coelicolor* A3(2), respectively, could synthesize various prenylated compounds from aromatic substrates, including flavonoids. The recombinant NphB and SCO7190 were overproduced in *E. coli* and purified to homogeneity. Results of the recombinant NphB catalysis are as follows: 7-*O*-Geranyl naringenin (**259**) and 6-geranyl naringenin (**260**) were synthetized from naringenin (**116**) and geranyl diphosphate (GPP); 7-*O*-geranyl apigenin (**261**) and 6-geranyl apigenin (**262**) were synthetized from apigenin (**135**) and GPP; 7-*O*-geranyl genistein (**263**) was synthetized from genistein (**2**) and GPP; 7-*O*-geranyl daidzein (**264**) and 8-geranyl daidzein (**265**) were synthetized from daidzein (**1**) and GPP. The result of the recombinant SCO7190 catalysis showed that 6-dimethylallyl naringenin (**266**) was synthetized from naringenin and DMAPP [90]. NphB, a propenyl transferase, can maintain regioselectivity for propenyl transfer in some cases.

A marine-derived *Streptomyces* sp. 060524 was able to hydrolyze the glycosidic bond of isoflavone glycosides. Genistein glycosides (**33**) and daidzein glycosides (**34**) were converted to genistein (**2**) and daidzein (**1**) by β-glucosidase, respectively. With the catalyzation by hydroxylase, the strain selectively hydroxylated at the 3′-position and transformed compounds **1** and **2** into 3′-hydroxygenistein (**16**) and 3′-hydroxydaidzein (**9**), respectively. In addition, glycitein glycoside (**37**) was also converted to glycitein (**267**) by β-glucosidase. Compounds **2** and **16** possessed excellent cytotoxicity against K562 human chronic leukemia [91].

A series of flavonoids incubating with the strain *Streptomyces* sp. M52104 could obtain transformational metabolites. Quercetin (**63**) was converted into quercetin-4′-*O*-β-D-glucuronide (**269**), quercetin-7-*O*-β-D-glucuronide (**270**), quercetin-3-*O*-β-D-glucuronide (**271**), and quercetin-3′-*O*-β-D-glucuronide (**272**) for 65 h of incubation. Furthermore, rutin (**268**) was only transformed into compounds **269-271** by incubating 90 h. Both 2*S*-naringenin (**273**) and 2*S*-naringenin-7-*O*-β-D-glucopyranoside (**274**) were able to convert into naringenin-7-*O*-β-D- glucuronide (**275**) and naringenin-4′-*O*-β-D-glucuronide (**276**) by incubating 65 h [92].

Quercetin **(63**) incubating with *Streptomyces rimosus* subsp. *rimosus* ATCC 10970 was able to be transformed into quercetin-7-*O*-β-4″-deoxy-hex-4″-enopyranosiduronic acid (**277**) in 72 h [93]. Biotransformation of genistein (**2**) in the culture of *Streptomyces* sp. MBT76 obtained 4′,7-di-hydroxy-5-methoxy-isoflavone (**278**), 4′-hydroxy-5,7-dimethoxy-isoflavone (**88**), 4′-hydroxy-5,7,8-trimethoxy-isoflavone (**279**), and 7,4′-dihydroxy-5,8-dimethoxy-isoflavone (**280**) [22]. However, methylated isoflavones exhibited lower bioactivity than the unmethylated precursors.

Two flavanones, pinocembrin (**54**) and naringenin **(116**), two flavones, chrysin and apigenin, together with a flavonol, kaempferol, were catalyzed by CYP105D7, a cytochrome P450 from *Streptomyces avermitilis* [94]. Through experiments, CYP105D7 enabled the catalysis of the hydroxylation of **54** and **116**. The 3′-position of **116** was hydroxylated by CYP105D7, while the hydroxylated position of **54** could not be determined. Nonetheless, the retention time detected in HPLC implied that 3-hydoxylation or 4′-hydoxylation did not occur.

A putative *O*-methyltransferase OMT gene SpOMT7740 was obtained from the sequence information of *Streptomyces peucetius* ATCC27952. Introducing the gene into *E. coli*, BL21 was used to detect the substrate promiscuity and perform functional characterization. Flavonoids were added into the fermentation broth of *E. coli* BL21 for biotransformation. The theoretical and experimental results were as follows: genistein (**2**), 3-hydroxyflavone (**98**), luteolin (**194**), 7,8-dihydroxyflavone (**281**), and phloretin (**282**) were successively converted into tri*-O*-methoxygenistein, 3-*O*-methoxyflavone, mono*-O*-methoxyleutolin, 7-hydroxy-8-*O*-methoxyflavone, and mono*-O*-methoxyphloretin [95].

Four known isoflavonoids, daidzein (**1**), genistein (**2**), daidzein-7-*O*-α-L-rhamnoside (**23**), and genistein-7-*O*-α-L-rhamnoside (**25**), were derived from Indonesian actinomycete *Streptomyces* sp. TPU1401A [32]. Compounds **1** and **2** were able to transform into the 7-*O*-glycoside derivatives, **23** and **25**, respectively, by the strain TPU1401A. Compounds **23** and **25** promoted the strain TPU1401A grown better than compounds **1** and **2**. This phenomenon suggested that the strain TPU1401A transformed isoflavones into isoflavone glycosides to hasten growth.

Studies showed that ring A hydroxyflavones were transformed to the corresponding C-4′ hydroxylated metabolites; the rate and yields of production were related to the distance between the C-4 carbonyl group and the hydroxyl group in the A ring. During the process of biotransformation in the culture of *Streptomyces fulvissimus*, 6-hydroxyflavone (**102**) and 7-hydroxyflavone (**121**) converted into 6,4′-dihydroxyflavone (**119**) and 7,4′-dihydroxyflavone (**122**) with a medium and slow reaction, respectively. 5-Hydroxyflavone (**158**) was completely transformed to 5,4′-dihydroxyflavone (**283**) and 5,3′,4′-dihydroxyflavone (**284**) for two days with a rapid reaction. These results demonstrated that the highest rate of transformation was observed when the hydroxy group was closest to the carbonyl group [96].

## 4. Development and Utilization of Flavonoids Synthesis Genes in *Streptomyces*

The availability of flavonoids in plants is limited by seasonal or regional variations, low abundance, and instability in the separation of single compounds from complex mixtures. Therefore, expressing plant biosynthesis genes in microorganisms is an effective method to produce these essential metabolites [97]. *Streptomyces*, one of the most important genera of actinomycetes, is diverse, and more than 50% of its species can produce various antibiotics and other active compounds of different structural types [98,99]. Enzymes with various catalytic functions in *Streptomyces* are useful members of an artificial gene cluster constructed in *E. coli* for the production of plant-specific flavones, including isoflavones and unnatural compounds by fermentation [100,101]. Here, we summarize the genes involved in flavonoid synthesis.

The most common method of studying genes that produce flavonoids in *Streptomyces* is the introduction of gene fragments into *E. coli* for heterologous expression. In plants, chalcone is a precursor of flavonoid, which can be converted to flavanones by chalcone isomerase, and 4-coumarate: coenzyme A ligase (4CL) is an essential enzyme in the transformation process. The gene fragment that synthesises 4CL has been found in *S. coelicolor* A3(2) and can be introduced into *E. coli* for heterogenic expression, which greatly enhances flavonoid production [100,102]. In a study by Kim et al., the SaOMT-2 gene from *S. avermitilis* MA-4680 was inserted into *E. coli* Sa-2 for expression, and the enzyme encoded by the SaOMT-2 gene was found to methylate the 7-hydroxyl group of isoflavones and flavonoids. In particular, the biotransformation of *E. coli* Sa-2 resulted in the production of sakuranetin, an antifungal flavonoid [103]. A gene fragment from *S. peucetius* ATCC 27952 that encodes an O-methyltransferase (OMT), SpOMT2884, was expressed in *E. coli*, and the enzyme was found to methylate various flavonoids, with 7,8-dihydroxyflavone being the most favourable substrate. Bioconversion of 7, 8-dihydroxyflavone was as high as 96% under optimal conditions, and the resulting 7-hydroxy-8-methoxyflavone was purified for in vitro glycosylation to produce glucose. The results indicated that methylation enhances the stability of the substrate, whereas glycation was shown to increase the water solubility of the substrate [104]. The SpOMT7740 gene from *S. peucetius* ATCC27952 encoding an OMT was cloned into *E. coli* to methylate different types of flavonoid substrates. The enzyme catalyzes the formation of various natural and non-natural *O*-methoxides [95]. *O*-Methylated phenylpropanoids are usually found in small quantities in plants. However, studies have indicated that co-culture of *E. coli* and *Streptomyces* can activate the expression of the SaOMT2 gene and promote the production of *O*-methylated phenylpropanoids, thus providing a powerful pathway for the production of scarce and valuable *O*-methylated phenylpropanoids [24]. Studies have shown that methylase GerMIII, derived from *Streptomyces* sp. KCTC 0041BP, can methylate the 4-hydroxyl position of some flavonoids, and quercetin may be the most suitable substrate [105]. The gene encoding oleandomycin glycosyltransferase (OleD GT) from *S. antibioticus* was introduced into *E. coli* BL21(DE3) for expression, and the purified recombinant OleD GT catalyzed glycosylation of various flavonoids. Fixation of OleD GT in hybrid nanoparticles of Fe_3_O_4_/silica/NiO is an effective technique for maintaining its high activity [106]. Not only *E. coli* but also *Streptomyces* can be used as engineered strains for heterologous expression of flavonoid-producing genes. In a study, flavonoid and stilbene biosynthesis genes were introduced for heterologous expression in *S. venezuelae* DHS2001, with the deletion of the native pikromycin polyketide synthesis gene, and the resulting strain produced racemic naringenin and pinocembrin from 4-coumaric acid and cinnamic acid, respectively. The yield of these two flavonoids was considerably increased by codon optimisation, and this is the first report of the phenylpropanoid biosynthesis pathway in the *Streptomyces* genus [107]. Marin et al. used the heterologous biosynthesis of industrial actinomyces *S. coelicolor* and *Streptomyces albus* to express the novo biosynthesis genes of three important flavonols, namely myricetin, kaempferol, and quercetin [108]. The shuffled biphenyl dioxygenase holoenzyme encoded by the *bphA1 (2072) A2A3A4* gene cluster of *S. lividans* exhibits a wide range of substrate specificity and hydroxylates several flavonoids and isoflavones; some of the obtained products possess free radical scavenging activity [89]. Prenyltransferase expressed by the prenyltransferase gene from *Streptomyces* prenyltransferase HypSc (SCO7190) was shown to exhibit extensive substrate specificity in the host plant, tomato, and therefore, tomatoes accumulate prenylated flavonoids. This is the first report of prenylated flavonoid accumulation in transgenic plants [109]. Through gene exploration of *S. clavuligerus*, three naringenin-producing genes, namely *ncs*, *ncyP,* and *tal,* were found, all of which are indispensable for the process. This is the first report on the natural production of naringenin in actinomycetes [110]. The queD gene from *S. eurythermus*^T^ encodes quercetinase (QueDHis6), which can convert several flavonols. The biological function of this enzyme was studied and was found to mainly play a role in detoxification rather than in catabolism [111]. The discovery of genes involved in the production of flavonoids by *Streptomyces* combined with newly developed techniques and increased genetic knowledge to manipulate the biosynthesis pathways is an effective way to achieve large-scale production of the essential flavonoids.

## 5. Conclusions

Flavonoids and isoflavonoids have attracted considerable attention due to their low toxicity and remarkable biological activity. Flavonoid and isoflavone pathways are considered the most characteristic natural product pathways in plants. However, microbial sources of isoflavones and flavonoids have received little attention. This is the first review of the fungal and actinomycete sources of isoflavones and flavonoids. Only 24 new compounds of isoflavones and flavonoids from fungi and actinomycetes have been identified; however, some of these are highly potent bioactive compounds that might be potential drug candidates. Compounds **61** and **58** from *Acanthostigmella* sp. CL12082 demonstrated potent antifungal activity. Compound **61** displayed strong antifungal activity against *C. albicans* and *A. fumigatus*, with IC_50_ values of 0.11 and 0.54 μg/mL, respectively. Similarly, compound **58** displayed extremely strong activity against the fungi, *C. albicans* and *A. fumigatus,* with IC_50_ values of 0.035 and 0.10 μg/mL, respectively [54]. Compound **76** from *Streptomyces* sp. (strain G246) showed broad-spectrum antimicrobial activity, and its antimicrobial activity was far superior than that of its positive controls, streptomycin and cycloheximide. Compound **76** displayed excellent inhibitory activity against *P. aeruginosa*, *S. enterica*, *E. faecalis*, *S. aureus*, *B. cereus*, and *C. albicans* with IC_50_ values of 16, 32, 8, 1, 4, and 8 μg/mL, respectively [60]. The new compounds **77**–**79,** derived from marine *Streptomyces* sp. G248, also demonstrated significant broad-spectrum antimicrobial activity against six pathogens, with IC_50_ values in the range of 1–16 μg/mL, and their activities were much higher than those of the positive controls, streptomycin and cycloheximide [61]. To ascertain the therapeutic potential of these compounds, particularly of lavandulylated and halogen atom-bearing flavonoids, further pharmacological, chemical, and toxicological studies are required. Nearly all the isoflavones and flavones from actinomycetes are reported to be produced by *Streptomyces* sp.; however, two studies have reported the isolation of isoflavones and flavones from actinomycetes, *M. aurantiaca* 110B and *Amycolatopsis* sp. [33,43]. The culture medium of *M. aurantiaca* 110B for producing compounds **20**, **21**, and **7****1** was investigated, and the results indicated that the compounds could be detected in the culture medium only when soybean cake is added to the medium. The results also indicated that *M. aurantiaca* 110B could transform plant daidzein into fucosylated derivatives. Notably, the fermentation medium in which *Amycolatopsis* sp. produces isoflavones also comprises of soybean meal. Whether these two strains have similar transformation pathways to produce isoflavonoid glycosides requires further investigation.

Because of the low bioavailability of isoflavones and flavonoids, the bioconversion of isoflavones and flavonoids has gradually gained research attention [80]. In this review, we summarised the biotransformation of isoflavones and flavonoids by fungi and actinomycetes. Understanding the biotransformation process helps us elucidate not only the metabolic pathways of these compounds but also their mechanisms of action, toxicity, and pharmacological activity [112,113,114]. *A. niger* and *Streptomyces* sp. are the most adaptable fungi and actinomycetes, respectively, and are the most commonly used biotransformation strains because of their ability to use various isoflavones and flavonoids as substrates for biosynthesis and biotransformation. They usually undergo hydroxylation, methylation, glycosylation, cyclisation, halogenation, and double bond reduction to produce isoflavones and flavonoids, which are more beneficial to human health; for instance, methylation enhances substrate stability, hydroxylation enhances antioxidant activity, and glycosylation enhances water solubility. The biotransformation processes in fungi and actinomycetes for producing isoflavone- and flavonoid-derived drugs have gained considerable attention.

Use of bacteria to produce plant-derived compounds is a new metabolic engineering technology that can help us synthesise important drugs and produce bioactive substances. Multiple flavonoid biosynthesis genes of *Streptomyces* sp. can be heterogeneously expressed in *E. coli* or optimisation of the gene expression strategy can be employed to obtain heterogenous expression strains. Moreover, transfer of the biosynthesis pathway from the original microorganism to a more compliant alien host may provide an effective platform for the production of desired levels of flavonoids or other novel compounds. To date, only flavonoid synthesis genes have been studied in *Streptomyces* sp., and no isoflavone synthesis gene has been discovered yet. Future research should focus on elucidating the generation mechanism of isoflavones and flavonoids in microorganisms to promote the utilisation of these compounds in the pharmaceutical field.

## Figures and Tables

**Figure 1 molecules-25-05112-f001:**
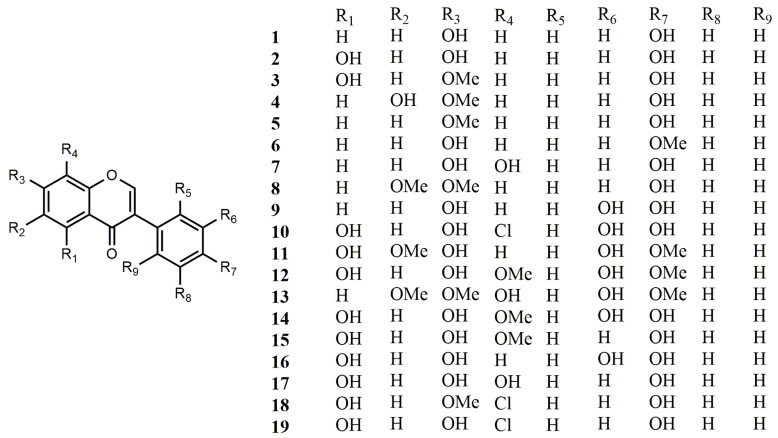
Reported structures of simple isoflavones **1**–**19** from fungi and actinomycetes.

**Figure 2 molecules-25-05112-f002:**
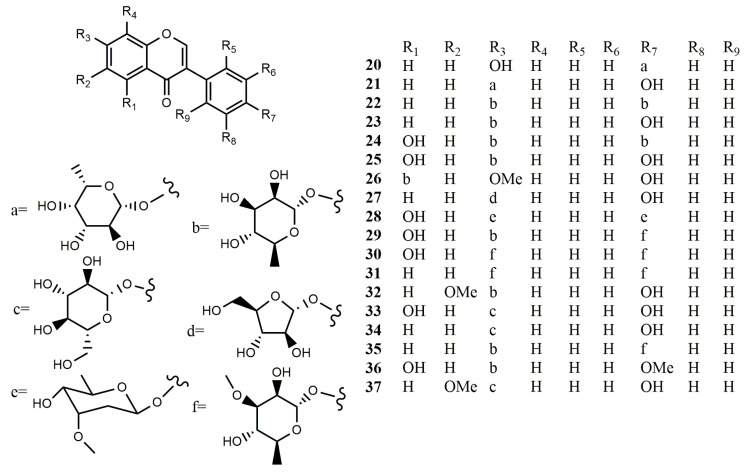
Reported structures of isoflavonoid glycosides **20**–**37** from fungi and actinomycetes.

**Figure 3 molecules-25-05112-f003:**
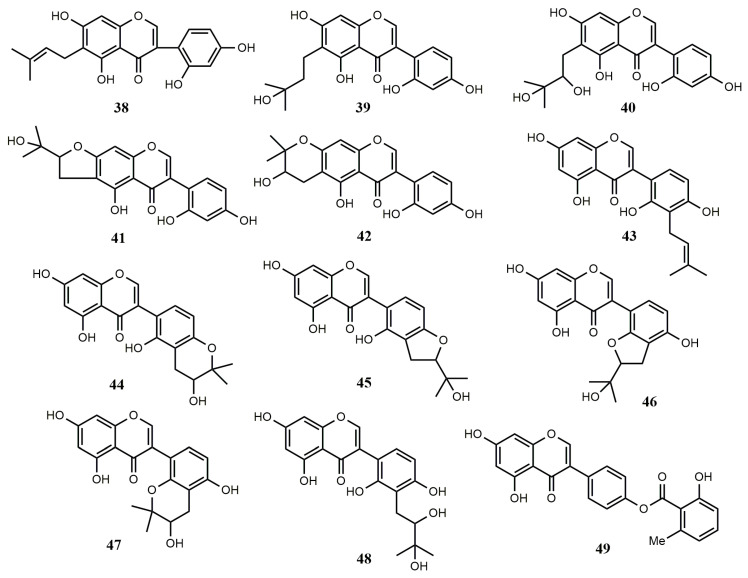
Reported structures of complex isoflavones **38**–**49** from fungi and actinomycetes.

**Figure 4 molecules-25-05112-f004:**
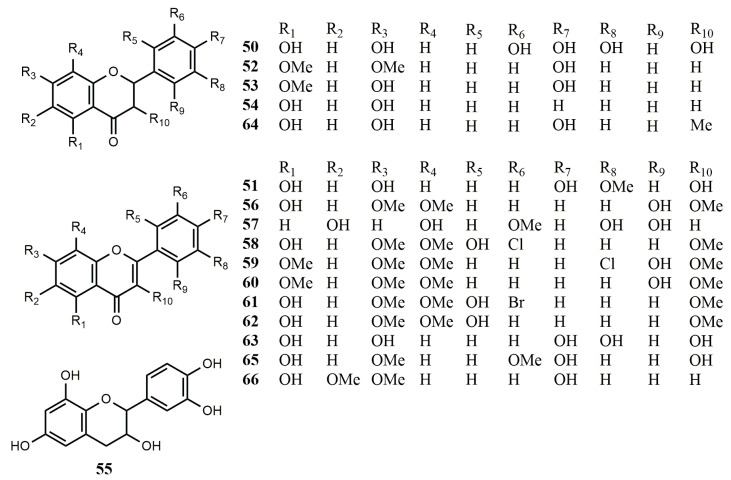
Reported structures of simple flavonoids **50**–**66** from fungi and actinomycetes.

**Figure 5 molecules-25-05112-f005:**
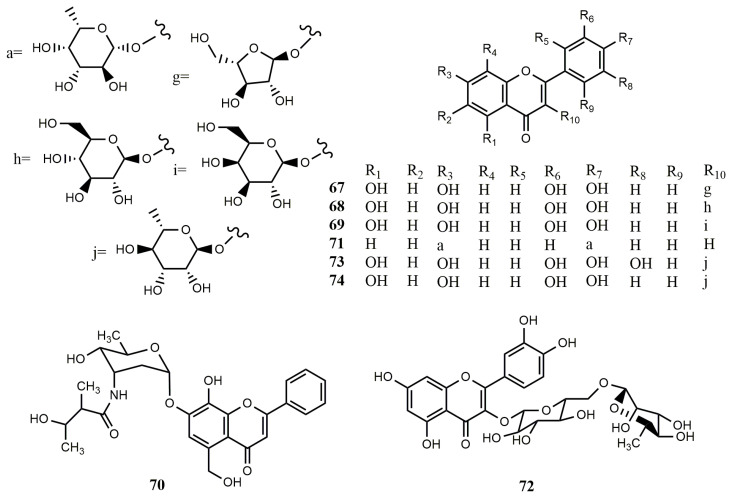
Reported structures of flavonoid glycosides **67**–**74** from fungi and actinomycetes.

**Figure 6 molecules-25-05112-f006:**
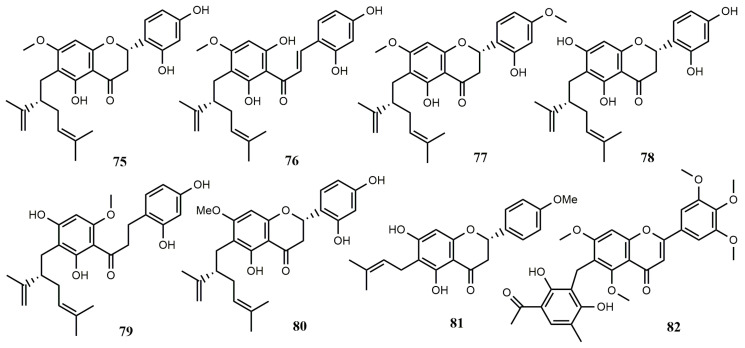
Reported structures of complex flavonoids **67**–**74** from fungi and actinomycetes.

**Figure 7 molecules-25-05112-f007:**
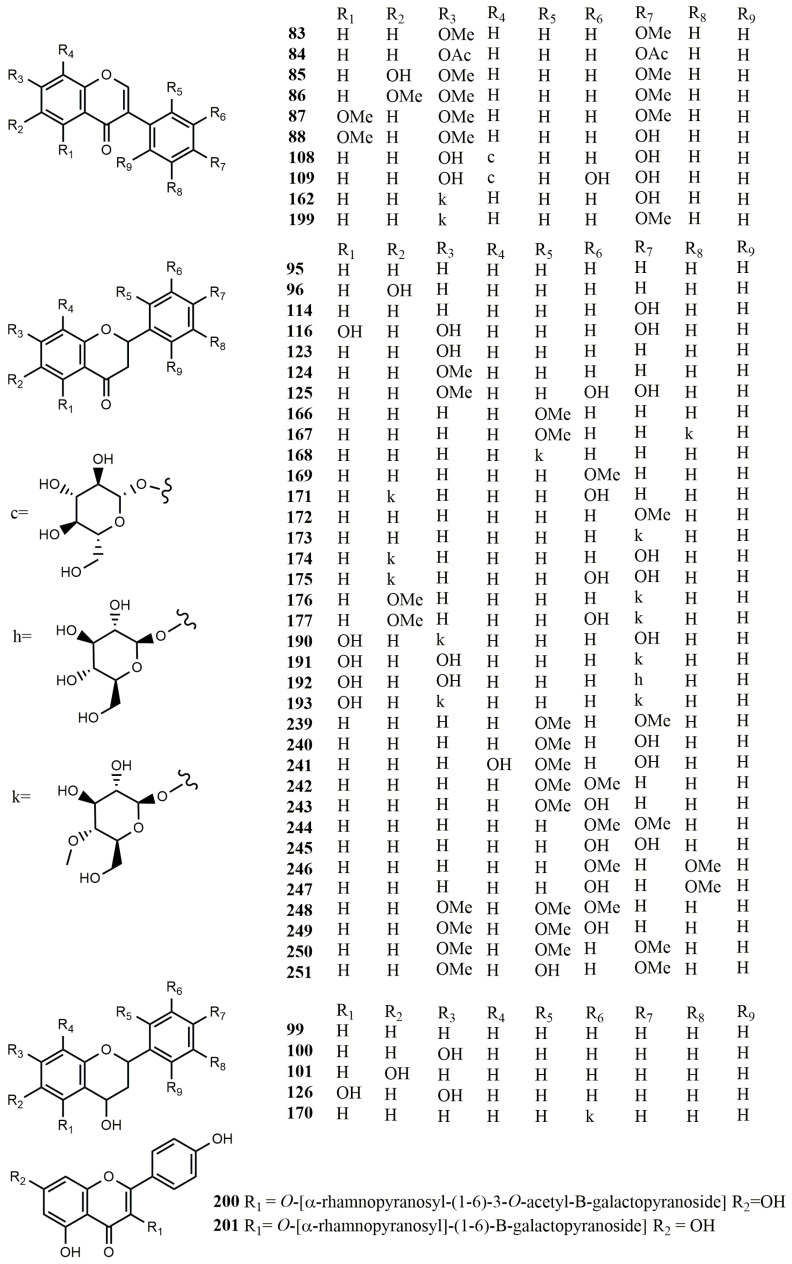
Biotransformation of isoflavones and flavonoids **83**–**251** by fungi.

**Figure 8 molecules-25-05112-f008:**
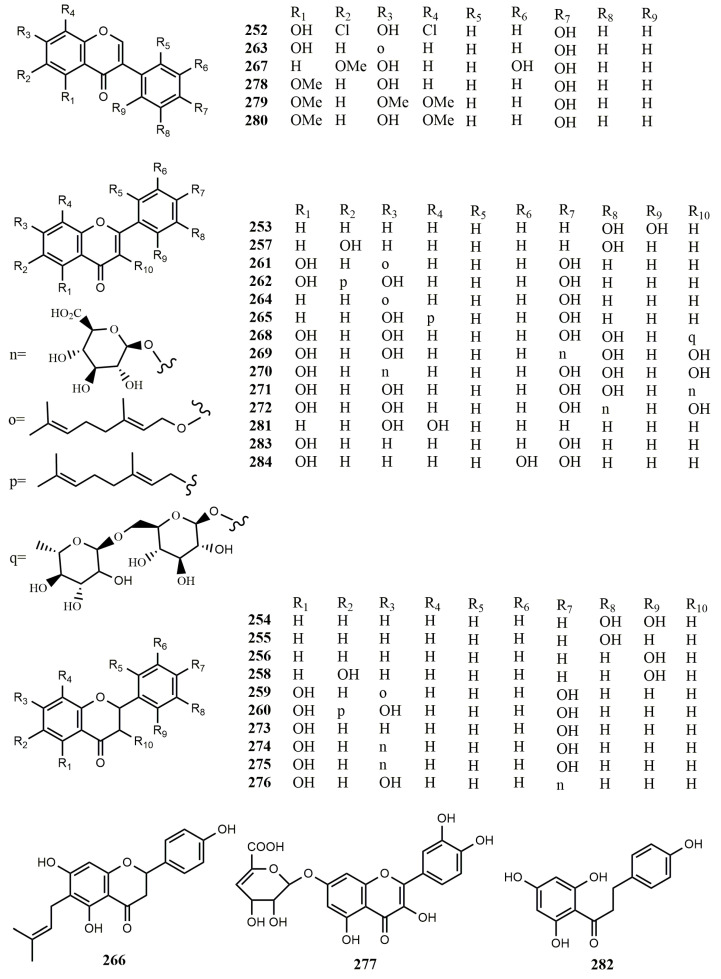
Biotransformation of isoflavones and flavonoids **252**–**284** by actinomycetes.

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
