# Peer review of "Naturally Occurring Flavonoids and Isoflavonoids and Their Microbial Transformation: A Review"

_molecules, 2020, doi:10.3390/molecules25215112_

Round 1
Reviewer 1 Report
Referees comments to the manuscript: Naturally occurring flavanoids and isoflavanoids and their microbial transformation: a review by Wang et al
The manuscript presents a review focused on microbial transformation of flavonoids and isoflavonoids.
Flavonoids are secondary metabolites of plants, playing multiple roles in physiology and ecology of individual plant species. Bacteria and fungi have developed metabolic tools during coevolution with plants that allow modification of phytoalexins of various chemical nature, including polyphenols. Bioactivity of these modified phenolics differ in multiple ways from that of original compounds, which is important not only for their interactions with the modifying organism. While the evolutionary goal of these modifications was to neutralize the defensive compounds (phytoalexins), eventually to get an additional nutrient, another consequence may be a significant modification of pharmacological activity, dietary value (e.g. inactivation of an antifeedant) etc. The manuscript of the review article by Wang et al is guided by an effort to collect and classify up to date knowledge of metabolic modifications of flavonoids and especially isoflavonoids. While an overview of selected original articles deserves credit, some inaccuracies or excessive simplifications in the introduction should be corrected or supplemented.
- While the title is Naturally occurring flavanoids and isoflavanoids…refers to flavanoids, i.e. compounds missing the keto group at C4, whole manuscript deals with flavonoids and isoflavonoids. This terminology should be unified.
- Flavonoids sensu lato include also isoflavonoids – this basic fact should not be overlooked. For classification of flavonoids see e.g. Ref. 1. From this point of view, the statements at rows 29-35 are misleading. Isoflavonoids originate from the same biochemical pathway as other flavonoids, i.e. combining the fenylpropanoid and the polyketide synthesis to yield chalcones which are then cyclized to flavanones. The fact, that isoflavones originate from flavanones (i.e. naringenin or liquiritigenin) by the activity of Isoflavone synthase, should be stated (see e. Ref 2,3,4).
- Section 2.1. Isoflavonoids – rows 57-58. The statement: "… isoflavones are mainly confined to the family Leguminosae [21,22]" is supported by two citations of a relatively older date (i.e. 1999). Since these times several comprehensive reviews on the occurrence of isoflavones in non-leguminous taxa have been published ( see e.g. ref. 5,6,7) as well as the review of newly discovered isoflavonoids in legumes (Ref 8, 9).
- I miss a clear and critical distinction between the complete synthesis of flavonoids in microorganisms and fungi (i.e. products of cultivation on flavonoid sensu lato free medium) and the ability to synthesize new (iso)flavonoids by modifying already existing (iso)flavonoid precursors. For example the rows 177-179: “The endophytic fungus Xylaria papulis BCRC 09F0222 was isolated from hairy woody plants. Three flavonoids, named myricetin (50), myricitrin (73), and quercitrin (74), were obtained from this strain [40].” However, according to the EXPERIMENTAL section of the cited article (DOI 10.1007/s10600-015-1327-3), the Xylaria papulis fungus had been cultivated on rice – a feeding plant known to produce myricetin and many other flavonoids (Ref. 10)
- Some of reported methylations are not sensu stricto modifications of flavonoids as the methylation occurred on the C4 hydroxyl of the sugar moiety of a (iso)flavonoid glycoside – see rows 423-424 and 431-432.
“Formononetin (6) was methylated to generate 4'-methoxyisoflavone 7-O-β-D-(4-O-methyl) glucopyranoside (199) by the strain ACCC 37814 [76]”
- Use italics, capitalization and spelling should be checked thoroughly. For example:
- Row 514 “hydrolze”
- Row 665 “and” should not be in italics
- 944 and 945 – “streptomyces” should be written Streptomyces
References
- Andersen M., Markham K.R.: Flavonoids, Chemistry, Biochemistry and Applications. CRC Press/Taylor & Francis, Boca Raton. 2006. ISBN 0-8493-2021-6.
- Akashi, T., Aoki, T., Ayabe, S., 1999. Cloning and functional expression of a cytochrome P450 cDNA encoding 2-hydroxyisoflavanone synthase involved in biosynthesis of the isoflavonoid skeleton in licorice. Plant Physiol. 121, 821–828.
- Overkamp, S., Hein, F., Barz, W., 2000. Cloning and characterization of eight cytochrome P450 cDNAs from chickpea (Cicer arietinum L.) cell suspension cultures. Plant Sci. 155, 101–108.
- Steele, C.L., Gijzen, M., Qutob, D., Dixon, R.A., 1999. Molecular characterization of the enzyme catalyzing the aryl migration reaction of isoflavonoid biosynthesis in soybean. Arch. Biochem. Biophys. 367, 146–150.
- Reynaud, J., Guilet, D., Terreux, R., Lussignol, M., Walchshofer, N., 2005. Isoflavonoids in non-leguminous families: an update. Nat. Prod. Rep. 22, 504–515
- Mackova, Z., Koblovska, R., Lapcik, O., 2006. Distribution of isoflavonoids in non-leguminous taxa – an update. Phytochemistry 67, 849– 855.
- Lapcik O. (2007) Isoflavonoids in non-leguminous taxa: A rarity or a rule? Phytochemistry 68: 2910-2916.
- Veitch, N. (2013). Isoflavonoids of the Leguminosae. Nat. Prod. Rep. 30, 988–1027.
- Al-Maharik N. (2019) Isolation of naturally occurring novel isoflavonoids: an update. Nat. Prod. Rep. 36, 1156-1195.
- Dey N., Bhattacherjee S.: Accumulation of Polyphenolic Compounds and Osmolytes under Dehydration Stress and Their Implication in Redox Regulation in Four Indigenous Aromatic Rice Cultivars. Rice Science, Vol. 27, No. 4, 2020
Reviewer 2 Report
This review summarizes the isoflavones and flavonoids derived from fungi and actinomycetes and describes the biotransformation of these substrates by the same microorganisms. Different biotransformation by actinomycetes and fungi are discussed, including hydroxylation, halogenation, methylation glycosylation, cyclization etc.
The review reads nicely in most parts, however the manuscript needs minor revisions in the following points:
- Line 43: …the presence isoflavonoids and flavonoids… -> the presence OF isoflavonoids and flavonoids…
- Line 189: the ref. [43] must not be superscript.
- Line 563: please correct the character of the ref. [89]
- Line 755, ref 26: the reference number is repeated twice
- Line 842: in the ref. 56 the journal name is missing, please add it (Bioscience, Biotechnology, and Biochemistry)
Reviewer 3 Report
In the present review the flavonoids and isoflavonoids derived from fungi and actinomycetes are summarized and classified and described their biological activities. Their biotransformation for the utilization of highly active biofunctional derivatives is described and assessed. There are also summarized the genes involved in flavonoid biosynthesis.
The article is particularly challenging for readers.
To the article, I have next comments and recommendations:
- In the whole article you mention flavonoids and isoflavonoids, why you have the names flavanoids and isoflavanoids in the titel of the paper, it is in this respect wrang, correct it!
- Letters D- and L- should be smaller as D- and L- . Correct it in the article.
- 125: correct some typing errors, like genistein; L. 148: 3,3-dimethylallyl; L.172: 72 h; L. 195: dechlorochlorfavon; L. 218: isoquercitrin; L. 287: 7-hyroxyflavan-4-ol; L. 298: …is 1.3 times higher than; L. 321: 3´,4´-dihydroxy-7-methoxyflavanone; L. 324: 2.4-cis-7-hydroxyflavan-4-ol : cis- and trans- should be in italics; L. 462-470: do not use dash dihydroxychalcone or trimethoxyflavanone, should be : 2´´-methoxy-2´,5´´-dihydroxychalcone, 2´´-methoxy-2´,3´-dihydroxychalcone, 2´,3´´,4´´-trihydroxychalcone, 5´-methoxy-2´,4´´-dihydroxychalcone, etc. Check it carefully!
- 510: synthetized
- 514: to hydrolyze…
- 546-548: should be: tri-O-methoxygenistein, 3-O-methoxyflavone, mono-O-methoxyluteolin, 7-hydroxy-8-O-methoxyflavone, mono-O-methoxyphloretin.
